# Comparison of Human Resources Management in Non-Family and Family Businesses: Case Study of the Czech Republic

**Petra Horváthová *, Marie Mikušová and Kateřina Kashi**

Department of Management, Faculty of Economics, VSB-Technical University of Ostrava, Sokolská třída 33, 702 00 Ostrava 1, Czech Republic; marie.mikusova@vsb.cz (M.M.); katerina.kashi@vsb.cz (K.K.)
* Correspondence: petra.horvathova@vsb.cz

**Abstract:** Human resources management, which includes a wide range of activities, may vary between businesses. One of the reasons for these differences may be the fact that they are non-family or family businesses. The purpose of this study is to identify differences in the area of human resources management between non-family and family businesses operating in the Czech business environment. The authors formulated three research questions and two hypotheses. The article's main findings are: non-family and family businesses do not substantially differ in human resources management. The article is formulating more general conclusions in the researched area, which can serve as a starting point for further directions of possible research on this issue.

**Keywords:** human resources management; non-family business; family business; sustainable business

## 1. Introduction

Sustainable entrepreneurship, based on the principles of sustainable development also includes social sustainability, expressed in personnel policy, which includes management and care for all employees. If the company fails to properly manage and maintain the employees, they may leave the company, which may have significant implications in very sensitive areas, such as business strategy, achievement of goals, company culture, or the morale of employees. The company may suffer significant financial losses by the departure of employees, as well as its economic (and thus environmental) sustainability and its existence may even be jeopardized. On the other hand, the well-managed, strategically linked and well executed management of all employees becomes a significant competitive advantage for successful companies and can contribute to its sustainable business.

The basic strategic objectives of each organization include long-term growth and sustainability [1]. In today's difficult conditions of global competition, and under the increasing pressures of the globalization of the business environment, human resources are becoming increasingly important for the achievement of these basic strategic objectives. The success of any business depends on people and their management which decide not only whether the business succeeds, but whether it will survive at all in the turbulent conditions of the contemporary world. Human resources management has fundamental influence on companies' sustainable businesses, which has to be considered as the priority of any business functioning. It is not financial resources, modern and efficient technology or well-developed strategy, but people—efficient employees—that play a crucial role in achieving performance and maintaining the stability of each company, and they are also the main competitive advantage for them [2]. Without qualified and knowledgeable employees, there is no possibility of any enterprise; thus their management becomes an essential component of its functioning [3]. Human resources management should be understood as a concept that forms part of the company management

that focuses on everything that is associated with the human factor in the work process. It is therefore decision-making in the area of human (employment) relations that affects the performance of workers and enterprises [4]. Human resources and their management play a critical role in the successful operation of any company [5]. Practical human resources management, which affects the well-being of employees and thus the performance of the company [6,7], and which includes a wide range of activities (e.g., development of personnel strategy and policy, ways of recruiting and selecting, evaluating, remunerating or promoting employees, etc.), may vary among businesses, depending on whether it is a non-family or family business. The importance of human resources management for the efficient functioning of businesses, both non-family and family, is reflected in scientific research; many of them also address the differences between the two types of business. Czech scientific literature also deals with this topic, especially the management of human resources in non-family enterprises. There is only a small amount of Czech research dedicated to family businesses.

The issue of family business is very important in terms of the development of the business environment and GDP creation. Family businesses are, therefore, crucial for economic progress [8,9].

On the basis of the above, the authors decided to contribute to the expansion and completion of the theoretical base in the relevant area by carrying out a survey aimed at mapping the differences in selected aspects of human resources management in non-family and family businesses in the Czech Republic. Based on the overall evaluation of the acquired data, the result of the survey is to answer the research questions, disprove/confirm formulated hypotheses and formulate more general conclusions in the research area as a basis for further directions of possible research for this issue. According to the authors and available facts, such a survey has not been conducted in the Czech Republic yet.

## 2. Literature Review

The management of human resources in non-family businesses is attended by a number of prominent authors (e.g., in [10–13]), and knowledge in this field is well known. However, the situation is different for family businesses. As mentioned above, as far as the Czech environment is concerned, only a limited number of professionals are devoted to family businesses. Therefore, the next chapter will mainly focus on family businesses.

The definition of a family business itself is not unambiguous—the definition of a family business varies from one country to another, but it always takes into account the number of family members participating in the company's capital, the amount of equity to be paid by family members, enterprise successor generation, etc. The most important feature is that the business is based on the personality of the founder, on the intensity of family involvement in the business, on the commitment of family members to perform more on their own and on quality, because to represent their own name and tradition is a prerequisite for success in future generations [14].

Many research articles focus on family businesses, many of which also address differences between non-family and family businesses. The differences are found mainly in the area of personnel and economics [15]. The family business combines more financial and non-financial objectives. Their managers aim to continue to the business across generations and to maximize the long-term value of the business [16]. The aim of non-family businesses, unlike family businesses, is primarily the return of investment by aligning the goals of the owners and managers, including their personal goals [17]. Craigh and Moores concluded, when examining business growth orientations, that family businesses are more conservative, risk averse and successors tend to conform to tradition [18]. Hubler identified differences resulting from the uniqueness of resources in family businesses, such as human and social capital, family financial capital and lower control costs [14]. There are specific attributes within a family business that have the potential to become both an advantage and a disadvantage. These bivalent attributes are systematically arranged in the model of Tagiuri and Davis [19]. Given the diversity of interests, norms, values and structure, the overlap of both systems—family and business—is dynamic with both positive and negative impacts, as shown by the Three Circle Model [19]. The specifics of the family business, which are due, among other things, to the intersection of contradictory factors,

are clearly presented in the Family Business Triangle model [20]. The model emphasizes the balance between the various dimensions of a family business, which are family and interpersonal relationships, business management and ownership. The model also implies the complexity of reconciling economic (e.g., profit, added value) and family (e.g., employment of family members) factors (e.g., in [21]). Factors that are very difficult to measure, such as devotion to family management or the intention to hand over a business to future generations, are used to determine the extent to which family goals are pursued [22]. This very limited list of specific characteristics of the family businesses, together with other features not mentioned here, that largely affect the human resources management area are examined in this paper.

The task of human resources management is to create conditions for increasing the intellectual capital of the company and creating a positive climate, which will be reflected in the economic performance of the company. Additionally, in this area of management, family businesses face additional complexities due to the integration of family, business and ownership systems. Flamini and Gnan see human resources management and family business performances as a duo that could receive more attention by scholars [23]. In fact, it is ascertained that high performing firms acknowledge the role of human resources management and they encourage the adoption and the implementation within family firms.

The positive impact of human resources management systems on firm performance has also been confirmed by the research of Gauci Borda et al. [24]. The findings show that there is a variety in the implementation of different combinations of human resources management practices on the spectrum between the control and commitment of human resources management in family businesses. Applying human resources management practices has an impact on the overall performance of the company, as does having precise work definitions and processes, which state the importance of sharing acquired knowledge and experiences with all of the company employees [25]. Efficiency, especially in family businesses, can be increased by motivating the employees, considering them not as machines but as human beings, satisfying their needs and integrating employees into the decision-making processes. Chopra emphasizes that companies will be more productive through utilizing an efficient human resources system, providing employees career planning, training and promotion opportunities [26].

In decision-making pertaining to human resources, many family business managers believe they have to choose between family and business [27]. The same authors state that there is a dearth of knowledge surrounding human resource practices and policies in the context of family businesses. They duly address this knowledge vacuum in presenting family business human resources strategies, supplemented by frameworks and tools for managing such strategies effectively. Additionally, they suggest that a thoughtful, holistic approach to human resources—and its continuous evolution—is a critical contributor to long-term success in family business, more so when compared to non-family business. According to these authors, the human resource practices in family business can be evaluated objectively. Family vision and values affect aspects of all components of the human resource cycle [28]. If the family and the company work together synergistically, the influence of the family will facilitate fair human resources management practices [9,29]. Koeberle-Schmid et al. emphasize the emotional component, which plays a significant role in the family company's human resources management, and the need to balance it with the rational side [30]. The link between socioemotional wealth and human resources management is also the subject of research by other authors (e.g., in [31–34]).

The disequilibrium of family and business interests also affect the process, methods and tools used in human resources management (e.g., in [35,36]). The ability of family business managers to adopt "best practices" in the area of human resources management is influenced by the cultural dimensions of the country where the business is located [37]. Benito-Hernández et al. show that a family nature of business has a positive impact in the use of human resources management external advice [38].

According to [38], the size of a family business brings further human resources management challenges. In addition to the classic institutional instruments in a business, such as the general meeting, the board of directors, etc., the family business also comprises other instruments, such as the family

protocol, family gatherings and more. The results of Başkurt and Altindağ show that the emphasis put on institutionalizing family businesses has a positive impact on employee performance and overall business performance [39].

Moreover, family businesses with a low level of formalization and weak governance practices can initially benefit from innovation and staff activity, but they must enhance their governance to sustain growth in performance [40]. The results of the research of Kidwell and Fish indicated that formalizing human resources systems occurs slowly in the industry [41]. The findings of Eddleston et al. lend some support to the argument that effective human resource practices in family firms should be balanced between instrumental governance mechanisms that reflect a monitoring approach and normative mechanisms that focus on collaborative efforts among family employees [42]. Family firms rely more on informal human resources practices, based on social networks. More formal human resources practices were found in family firms with a family CEO [43]. Michiels's research outputs are different: the results support the hypothesis that family firms with a family CEO adopt significantly less formal compensation practices than their counterparts, which are led by a non-family CEO [44].

Talent management also faces additional challenges in the family business, for example by objectively providing competent company staff and suppressing nepotism. Nepotism in the selection and placement of employees is the major problem within strategic human resources management in family businesses [39]. Bhala and Bratton formulated principles that, beyond these generally accepted best practices in talent management, can overcome these unique challenges [45]. The results from Springer and Hadrys-Nowak indicate that family businesses are doing slightly better with the implementation of TM processes than non-family Polish companies [46]. This difference, however, is relatively small. The specific family business talent management approach, which is embedded in family business culture is shaped by family values and emotional attachment [47].

Human resources management can play a role in preserving family business at a risky stage of succession [48]. Human resources management can help to diminish the risk of family business succession by reducing the dependence on family management through personnel empowerment. However, as Yedderm argues, potential positive human resources management outcomes for succession are only possible given the conditions of human resources management professionalization and family management support [48].

In their research, the authors focused only on one of the two basic areas of differences between Czech non-family and family businesses, namely personnel management. The aim of the article is to identify differences in the area of human resources management between non-family and family businesses operating in the Czech business environment. In this context, the authors formulated three research questions and two hypotheses:

(1) Is the overall concept of human resources management different for non-family and family businesses? The authors intended to find out whether the unique nature of family businesses (when more employee care, family access to employees, the influence of socioemotional wealth on the whole human resources management process, etc. are often declared) affects the overall concept of human resources management. Thus, to what extent, if any, is the concept of human resources management in non-family businesses different from family businesses.

(2) Do non-family businesses use tools other than family businesses within the survey population for active work with people (e.g., training, development, motivation, remuneration, etc.)? Of course, non-family members are employed in family businesses, often even in key positions. There may be situations where a non-family employee may feel injustice. Does the family character of the business influence the choice of tools for the selection of workers, their training, motivation, remuneration, etc.? Are family businesses trying for equal access to family and non-family employees? Therefore, does a family business prefer other tools than the non-family type?

(3) Does the informal approach in the field of human resources management prevail in family business? In the analyzed literature, researchers often encountered outputs that confirmed a rather informal approach to decision-making and problem solving, not only human resources

management-related problems. The authors wanted to confirm or refute the informal approach in human resources management policy in this research as well. The hypothesis was formulated in this sense, so H1: The informal approach in the human resources management system/concept is accepted to a greater extent in family businesses than in non-family businesses. However, at the same time, the authors considered that, even in family businesses, their managers are aware of the need for a formal approach. In this context, they created another hypothesis on this research question. They intended to disprove/confirm the idea that, even if the informal concept of human resources management prevails, a formalized approach will be preferred in some areas. The authors focused on the area of performance evaluation, as it is closely linked to remuneration and, therefore, closely related to costs, therefore H2: More than two thirds (i.e., 66% of companies in which the informal concept of human resources management is preferred), use formal evaluation of employees. Hypothesis H2 will be tested for family and non-family businesses.

The goal of the manuscript will be met both by evaluating the data obtained by the online questionnaire survey (thus answering the first two research questions) and by testing the formulated hypotheses (thus answering the third research question).

## 3. Methods and Data

A survey was conducted to investigate human resources management in non-family and family manufacturing industries in the Moravian–Silesian region. Its aim was to identify differences in two basic areas of human resources management—in the overall concept of human resources management and in the used human resources management tools in specific areas of human resources management.

Primary data were collected through a quantitative survey (using a questionnaire). This method was chosen for its advantages over the other options for addressing respondents. The advantages of this method are the ability to address a large number of respondents and, at the same time, address those who are not willing to provide a personal interview, as well eliminate the relationship between respondent and researcher, which may allow for obtaining a relatively exact and objective opinion. However, the questions must be precise and comprehensible so that they are clearly understood and do not require further explanation. Other advantages include, for example, low costs, high return, speed of answers, versatility and greater sincerity of respondents than, for example, in personal interviews. On the contrary, the risks of this method include the fact that not everyone is connected to the Internet and also susceptibility to technological problems [49].

The formulation of individual questions was verified by the Focus Group qualitative research method with the participation of 20 experts (10 non-family representatives and 10 family business representatives) from practice. It was determined whether the chosen method of online questionnaire survey was appropriate, how focus group participants responded to the questions of the questionnaire, whether the questions were well formulated, whether they understood the questions and whether the offered answering options were appropriate. Based on the outputs of the focus group, two questions were combined into one question, one dealing with the existence of a human resources strategy and the other with the existence of a human resources policy. The focus group participants agreed that they see the strategy as focused on the personnel area as a whole, and the policy as the rules that the company follows in any decisions that in some way directly or indirectly affect workers and work, and that in both areas they can be included in a single question that offers answers that cover the existing reality in practice. A similar situation occurred on two issues—one concerning the knowledge of the business owner and the skills of human resources management. They understood skills as the practical application of knowledge of this issue, and to ask these two things in one question, they argued that it is not possible to have skills without knowledge. Furthermore, the focus group participants requested that a question concerning the form of the personnel information system be included in the questionnaire. The result of the focus group meeting was the final questionnaire, which was later validated by experts from practice—the experts also commented on the extent to which individual questions and proposed answers described individual areas. Out of the total number of 15 questions,

only one did not differ much in the number of experts who considered it essential and those who considered it insignificant and useless. The content validity was also evaluated using the Content Validity Ratio (CVR). CVR = $[(E-(N/2))/(N/2)]$, where N is the total number of experts and E is the number of those who rated the object as essential. The final form of the questionnaire consists of 3 main topics and 15 items. The content validity ratio (CVR) ranged between 0.85 and 1.00 for each topic and from 0.75 to 1.00 for each item. Therefore, the questionnaire is considered valid from a quantitative point of view in terms of content. The final questionnaire was then used to survey human resources management in non-family and family business.

In the cover letter, respondents were informed of the purpose of the questionnaire survey, they were asked to complete the questionnaire and were also informed of the possibility of getting informed about the results of the survey. The questionnaire included one identification question (number of employees) and 15 questions concerning the subject of the survey, where respondents chose only one option from predefined answers. Issues covered by the first area related to human resources management strategy and policy, knowledge and skills in the area, conceptual and administrative aspects of human resources management, personnel activities, human resources management decisions and the personnel information system. The second area concerned the tools used in the areas of recruitment, selection, placement, training and development, motivation and retention of workers in the company, their performance appraisal, their remuneration and succession management (talent management).

The population consisted of all (more than 100,000) enterprises operating, according to CZ-NACE, in the manufacturing industry of the Czech Republic, divided according to Commission Regulation (EC) No 800/2008 (excluding independence) into a micro enterprise (<10 employees and at least one aspect: annual turnover < EUR 2 million and/or assets < EUR 2 million), small business (<50 employees and at least one aspect: annual turnover < EUR 10 million and/or assets < EUR 10 million) and medium-sized enterprise (<250 employees and at least one aspect: annual turnover < EUR 50 million and/or assets < EUR 43 million). This division of respondents was executed because of the fact that, in the future, it is envisaged to address businesses from other European countries and, therefore, a certain official view is needed in all EU countries. The sample included 12,632 enterprises, according to the structure shown in Table 1, selected using the random simple sampling technique. The original intention was to select 3000 businesses in each category ("size" and non-family/family), but this proved unfeasible for the following reasons: (1) more than one third of the businesses did not have available turnover or number information for employees (this is a specificity of the Czech Republic, as many companies do not submit complete financial statements—only enterprises with available turnover information have been included in the selection); (2) the European classification of enterprise size is considerably overvalued and the number of enterprises is falling drastically, much more steeply than in the EU in general, meaning, in practice, that there are many micro-enterprises in the Czech Republic, but far less SMEs; (3) the original estimate of the number of family businesses was 10% (i.e., 10,000) of the entire portfolio of companies in the Czech Republic; however, as it turned out, in the manufacturing industry the number of family businesses is smaller (about 6%), which can be seen as a specific segment.

**Table 1.** Structure of the data sample.

| | Non-Family Businesses | | Family Businesses | | Total | |
|---|---|---|---|---|---|---|
| | Absolute Frequency | Relative Frequency | Absolute Frequency | Relative Frequency | Absolute Frequency | Relative Frequency |
| Micro enterprise | 3000 | 36.4% | 2263 | 51.5% | 5263 | 41.6% |
| Small enterprise | 3000 | 36.4% | 1598 | 36.4% | 4598 | 36.4% |
| Medium enterprise | 2236 | 27.2% | 535 | 12.1% | 2771 | 22.0% |
| Total | 8236 | 100.0% | 4396 | 100.0% | 12,632 | 100.0% |

Source: own elaboration based on data from IBM SPSS Statistics 23.0 (2019).

The identification of family businesses was slightly problematic because there is no official database. At the time of the research, the family business in the Czech Republic was not yet defined. The starting point for identifying family businesses is the general definition of family business, which includes the aspect of the share of family members in the capital and the participation of family members in the management and control of the enterprise. For the purposes of this research, the authors decided how family businesses are understood. At least one of the following conditions had to apply to the survey: at least two persons with the same surname are among the owners of the enterprise and/or are in the executive or supervisory bodies of the enterprise. For the purposes of the survey, a specialized selection was made from the Czech Companies Database, which identified family businesses, according to the specified criteria. The authors admit that the used definition of a family business is restrictive.

Of the total of 12,632 questionnaires sent, 572 questionnaires were returned (288 from non-family and 284 from family-owned enterprises), with a return of 4.6%. For the detailed structure of respondents see Table 2.

**Table 2.** Structure of respondents.

| | Non-Family Businesses | | Family Businesses | | Total | |
|---|---|---|---|---|---|---|
| | Absolute Frequency | Relative Frequency | Absolute Frequency | Relative Frequency | Absolute Frequency | Relative Frequency |
| **Micro enterprise** | 86 | 29.9% | 96 | 33.8% | 182 | 31.8% |
| **Small enterprise** | 86 | 29.9% | 138 | 48.6% | 224 | 39.2% |
| **Medium enterprise** | 116 | 40.2% | 50 | 17.6% | 166 | 29.0% |
| **Total** | 288 | 100.0% | 284 | 100.0% | 572 | 100.0% |

Source: own elaboration based on data from IBM SPSS Statistics 23.0 (2019).

In the first phase of processing the obtained data through IBM SPSS Statistics 23.0, the authors focused on comparing non-family and family businesses only in selected areas of human resources management as a whole. The authors did not execute a detailed analysis of the results pertaining to the "size" of the business (i.e., micro enterprise, small enterprise, medium enterprise). This detailed analysis will be elaborated in other works of the authors, where the second stage will be sorted according to the size of the enterprise.

A two-sample comparison test was used to test the hypothesis of the informality of the human resources management concept—the informal approach in the human resource system/concept is accepted to a greater extent by family businesses than by non-family businesses. The numbers of both samples is higher than 50 and the sample proportions do not reach significantly small or large values. If the null hypothesis is valid, it can be assumed that the statistic test has an asymptotically normalized normal Z distribution.

$$Z = \frac{p_1 - p_2}{\sqrt{p^* \times (1 - p^*) \times \left(\frac{1}{n_1} + \frac{1}{n_2}\right)}},$$

$$p^* = \frac{n_1 p_1 + n_2 p_2}{n_1 + n_2},$$

where

$n_{1,2}$—sample size (i.e., amount of all answers),

$p_{1,2}$—relative frequency of positive answers (i.e., answers that prefer informal approach).

For non/confirmation of the assumption that more than 66% of companies with an informal human resources management concept will still formally evaluate the performance of their employees (regardless of whether they are family business or non-family business), a test of assumptions was

used. Calculations were performed at the level of significance $\alpha = 0.05$ when the value of normal distribution u$\alpha$ = 1.96, u0.05 = 1.96:

$$u = \frac{\frac{m}{n} - p_0}{\sqrt{\frac{p_0(1-p_0)}{n}}}.$$

N—number of organizations that accept informal human resources management concept,
M—number of organizations that use formal tool forms for employees' performance appraisals,
U—tested criteria for test of relative frequency.

## 4. Results and Discussion

In accordance with the objective of the article, the results are structured into three basic areas: (1) the overall concept of human resources management; (2) the tools used to manage human resources in specific areas of work with people; (3) the predominant approach in the field of human resources management in family business.

### 4.1. The Overall Concept of Human Resources Management

In the first part of the survey, which focused on the overall concept of human resources management in an enterprise, the respondents were asked seven questions, shown in Table 3:

**Table 3.** Questions about the overall concept of human resources management.

| Questions about the Overall Concept of Human Resources Management |
| --- |
| Do you have a human resources management strategy and policy in your company? |
| Does the business owner(s) have the necessary knowledge and skills in HRM? |
| Who provides the conceptual part of human resources management in your company? |
| Who manages the administration of human resources in your company? |
| Do you have a well-thought-out and prepared concept of personnel activities in your company at all times? |
| How are HR decisions in your company mostly made? |
| What form does the personnel information system have? |

The answers (those with the highest frequency only; nevertheless, all answers to all questions from the highest to the lowest frequency were processed) from 288 non-family and 284 family businesses are synoptically given in Table 4.

**Table 4.** The three most common characteristics of the individual areas of the overall human resources management.

| | Non-Family Businesses | Family Businesses |
| --- | --- | --- |
| **Strategy and policy** | yes, both, but not in a written form (86; 29.9%) | neither (128; 45.1%) |
| **Knowledge and skills** | yes, partially (188; 65.3%) | yes, partially (180; 63.4%) |
| **Conceptual part** | owner/owners (192; 66.7%) | owner/owners (200; 70.4%) |
| **Administration** | personnel department/personnel manager (124; 43.1%) | owner/owners (118; 41.6%) |
| **Concept of personnel activities** | yes, but only some (112; 38.9%) | yes, but only some (106; 37.3%) |
| **Decisions** | centralized, consultative (transparent) (132; 45.8%) | centralized, consultative (transparent) (160; 56.3%) |
| **Personnel information system** | none, we use other tools (166; 57.3%) | none, we use other tools (208; 73.2%) |

Source: own elaboration based on data from IBM SPSS Statistics 23.0 (2019).

Illustratively, the three most common characteristics of the individual areas of the overall human resources management concept are presented in Table 5.

**Table 5.** The three most common characteristics of individual areas of the overall concept of human resources management.

| Area | Non-Family Businesses | Family Businesses |
|---|---|---|
| **Strategy and policy** | yes, both, but not in a written form (86; 29.9%) | neither (128; 45.1%) |
| | neither (78; 27%) | yes, both, but not in a written form (68; 23.9%) |
| | both, in a written form (74; 25.7%) | both, in a written form (40; 14.1%) |
| **Knowledge and skills** | yes, partially (188; 62.5%) | yes, partially (180; 63.4%) |
| | yes, all of them (80; 27.8%) | yes, all of them (82; 28.9%) |
| | no (20; 6.9%) | no (22; 7.8%) |
| **Conceptual part** | owner/owners (192; 66.7%) | owner/owners (200; 70.4%) |
| | small group of managers (48; 16.7%) | small group of managers (40; 14.1%) |
| | senior manager (30; 10.4%) | senior manager (32; 11.3%) |
| **Administration** | personnel department/personnel manager (124; 43.1%) | owner/owners (118; 41.6%) |
| | owner/owners (84; 29.2%) | personnel department/personnel manager (80; 28.2%) |
| | small group of managers (28; 9.7%) | consulting or another specialized firm (30; 10.6%) |
| **Concept of personnel activities** | yes, but only some (112; 38.9%) | yes, but only some (106; 37.3%) |
| | yes, all of them (110; 38.2%) | yes, all of them (90; 31.7%) |
| | no (66; 22.9%) | no (88; 31%) |
| **Decisions** | centralized, consultative (transparent) (132; 45.8%) | centralized, consultative (transparent) (160; 56.3%) |
| | centrally, authoritatively ("behind closed doors", non-transparent) (96; 33.3%) | centrally, authoritatively ("behind closed doors", non-transparent) (90; 31.7%) |
| | collectively (participative) (60; 20.8%) | collectively (participative) (34; 12%) |
| **Personnel information system** | none, we use other tools (166; 57.6%) | none, we use other tools (208; 73.2%) |
| | higher-level information system (e.g., a comprehensive corporate information system (84; 29.2%) | higher-level information system (e.g., a comprehensive corporate information system (54; 19%) |
| | separate information processing system in the area of personnel (which, however, usually does not appear separately, but it is an integral part of enterprise information systems (38; 13%) | separate information processing system in the area of personnel (which, however, usually does not appear separately, but it is an integral part of enterprise information systems (22; 7.8%) |

Source: own elaboration based on data from IBM SPSS Statistics 23.0 (2019).

It can be stated that there are only two areas—the development of a human resources management strategy and policy and the administration of human resources management. While non-family businesses, in most cases (i.e., 86 (29.9%)) have a strategy and policy, but not in writing, family businesses, in most cases (i.e., 128 (45.1%)), have neither a strategy nor a human resources management policy. In the case of the second most common characteristic, the situation is reversed—while non-family businesses do not have a human resources strategy or policy in 78 (27.1%) cases, family businesses have a strategy and policy in 68 (23.9%) cases, however, in writing form. The administrative aspect of human resources management in non-family enterprises is, in most cases (i.e., in 124 (43.1%)), ensured by the HR department/personnel officer and in family enterprises, in most cases (i.e., in 118 (41.6%)), by the owner/owners. In the case of the second most common characteristic, the situation is again reversed, where, in non-family businesses, the administration/administration is provided in 84 (29.2%) cases by the owner(s). Regarding the third most common characteristic, the situation is different here—in

non-family businesses, the administration is ensured in 28 (9.7%) cases by a small group of managers, while, in family businesses, in 30 (10.6%) cases it is by consulting or another specialized firm.

Based on the aforementioned data, the answer to research question (1) is: The overall concept of human resources management is not basically different for non-family and family businesses; differences can only be found in two aspects out of seven—in human resources management strategy and policy design and in the administration of human resources management. This conclusion can be considered as a starting point for further directions of possible research on this issue (e.g., the possibility to identify the causes and consequences of the absence of human resources management strategy and policy for family businesses), to investigate the reasons and quality information systems both in non-family and family businesses.

Since these are collectively micro, small and medium-sized enterprises, the answers to most questions, with the exception of the well-thought-out and prepared conception of human resources activities, are in line with the authors' expectations, based on their own experience. All personnel activities need to be carried out in all companies, regardless of their size. Differences exist only in the scope and frequency of individual personnel activities. While large enterprises carry out staffing activities on a continuous basis, in micro, small and medium-sized enterprises, some staffing activities are only occasionally carried out. Some of them do not even have to be carried out for several years (e.g., activities related to the movement of workers to and from the enterprise). On the other hand, there is a constant need to pay attention to, for example, the appraisal of workers and their remuneration, and to care for them or their working relationships [3]. Micro enterprise, alongside the SMEs, must have a well-thought-out and prepared concept of all personnel activities at all times, and these enterprises must be able to activate any activity at any time and carry it out without undue delay, groping and improvisation.

On the basis of the latest theoretical knowledge of human resources management, practical experience of the authors and the results of the research focused on the first area (i.e., the overall concept of human resources management for both non-family and family businesses), some general recommendations can be formulated in this area. For micro, small and medium-sized enterprises: formulate strategy and policy in writing as well; ensure a full range of knowledge and skills of the owner(s) in human resources management; entrust the administrative part of human resources management to a person other than the owner(s); rethink and prepare a concept for all personnel activities; use the personnel information system.

*4.2. Human Resources Management Tools Used in Specific Areas of Work with People*

In the second part of the survey, which focused on the human resources management tools used in specific areas of working with employees, the respondents were asked eight questions, shown in Table 6.

The answers (only those with the highest frequency; nevertheless, all answers to all questions from the highest to the lowest frequency were processed) from 288 non-family and 284 family businesses are given in Table 7. It can be stated that differences exist in one aspect only.

**Table 6.** Questions about human resources management tools used in specific areas of work with people.

| Questions about Human Resources Management Tools |
| --- |
| Which of the human resources management tools in the area of acquiring employees do you use in your company? |
| Which of the human resources management tools in the area of selecting employees are you using in your company? |
| Which of the human resources management tools do you use to place employees in your company? |
| Which human resources management tools in employees' training and development do you use in your company? |
| Which human resources management tools do you use to motivate and retain employees in your company? |
| Which of the human resources management tools do you use in the field of the evaluation of the work performance of employees in the company? |
| Which of the human resources management tools in the area of employee remuneration do you use? |
| Do you manage succession in your company (do you apply talent management)? |

**Table 7.** Most used human resources management tools in individual areas.

| | Non-Family Businesses | Family Businesses |
| --- | --- | --- |
| **Acquiring** | potential employees offer themselves (150; 52.0%) | potential employees offer themselves (130; 45.7%) |
| **Selection** | curriculum vitae (140; 48.6%) | curriculum vitae (130; 45.7%) |
| **Placement** | aligning the employee's profile (242; 84.0%) | aligning the employee's profile (216; 76.1%) |
| **Training and development** | coaching (102; 35.4%) | coaching (96; 33.8%) |
| **Motivation and retention** | appropriate reward (250; 86.8%) | appropriate reward (246; 86.6%) |
| **Performance management/appraisal** | informal (continuous) appraisal (142; 49.3%) | informal (continuous) appraisal (136; 47.8%) |
| **Remuneration** | basic salary (236; 81.9%) | basic salary (228; 80.2%) |
| **Succession management** | no, but we are considering introducing a formal system (112; 38.8%) | we manage succession based on family or personal relationships (118; 41.5%) |

Source: own elaboration based on data from IBM SPSS Statistics 23.0 (2019).

The three most common human resources management tools in each area are illustrated in Table 8. It can be stated that differences exist only in three areas. There are minor differences in the areas of training and development (differences in the second- and third most used tools) and in the area of motivation and retention (differences in the third most used tool; moreover, the frequency of using these tools is very low). While the second most used tool in training and development is project work for non-family businesses (82; 28.5%), for family businesses it is the task assigned (80; 17.6%), and for non-family businesses, mentoring (40; 13.9%), whereas for family businesses, family or personal relationships (44; 15.5%). In the area of the motivation and retention of employees, the third most used tool for non-family enterprises is healthy working and interpersonal relationships in the workplace (6; 2.1%), whereas for family enterprises it is two tools—family or personal relationships (8; 2.8%) and training and development (8; 2.8%). Greater differences exist in the two most commonly used tools in the field of succession management. While non-family businesses most often do not manage succession but consider introducing a formal system (112; 38.9%), for family businesses, succession is managed on the basis of family or personal relationships (118; 41.6%). The second most common tool for non-family businesses is family-based (or rather) personal relationship management (84; 29.2%),

while family businesses, without considering a formal system, are operative solutions to succession (74; 26.1%).

**Table 8.** The three most used human resources management tools in individual areas.

| Area | Non-Family Businesses | Family Businesses |
|---|---|---|
| **Acquiring** | potential employees offer themselves (150; 52.0%) | potential employees offer themselves (130; 45.8%) |
| | new employee is recommended by current employee of the company (62; 21.5%) | new employee is recommended by current employee of the company (66; 23%) |
| | businesses directly address the chosen individual (28; 9.7%) | businesses directly address the chosen individual (32; 11.3%) |
| **Selection** | CV (140; 48.6%) | CV (130; 45.8%) |
| | questionnaire (78; 27.0%) | job interview (68; 23.9%) |
| | job interview (48; 16.7%) | questionnaire (50; 17.6%) |
| **Placement** | aligning the employee's profile with the job vacancy (242; 84.0%) | aligning the employee's profile with the job vacancy (216; 76.1%) |
| | tailoring (22; 7.6%) | family or personal relations (38; 13.4%) |
| | family or personal relations (20; 6.9%) | tailoring (18; 6.3%) |
| **Training and development** | coaching (102; 35.4%) | coaching (96; 33.8%) |
| | work on projects (82; 28.5%) | task delegation (50; 17.6%) |
| | mentoring (40; 13.9%) | family or personal relations (44; 15.5%) |
| **Motivation and retention** | appropriate reward (250; 86.8%) | appropriate reward (246; 86.6%) |
| | interesting and appreciated work (22; 7.6%) | interesting and appreciated work (14; 4.9%) |
| | healthy working and interpersonal relationships in the workplace (6; 2.1%) | family or personal relations (8; 2.8%) training and development (8; 2.8%) |
| **Performance management/ appraisal** | informal (continuous) appraisal (142; 49.3%) | informal (continuous) appraisal (136; 47.9%) |
| | formal (systematic) appraisal (130; 45.1%) | formal (systematic) appraisal (124; 43.7%) |
| | family or personal relations (8; 2.8%) | family or personal relations (16; 5.6%) |
| **Remuneration** | basic salary (236; 81.9%) | basic salary (228; 80.3%) |
| | short-term incentives and bonuses (26; 9.0%) | short-term incentives and bonuses (22; 7.8%) |
| | optional premiums (10; 3.5%) | optional premiums (14; 4.9%) |
| **Succession management** | no, but we are considering introducing a formal system (112; 38.9%) | based on family or personal relationships (118; 41.6%) |
| | based on family or personal relationships (84; 29.2%) | operatively, we do not consider establishing a formal system (74; 26.1%) |
| | rather informal and intuitive character deriving from personal interactions and leadership style (76; 26.4%) | rather informal and intuitive character deriving from personal interactions and leadership style (58; 20.4%) |

Source: own elaboration based on data from IBM SPSS Statistics 23.0 (2019).

On the basis of the aforementioned data, the answer to research question (2) is: The tools used in specific areas of work with people are hardly different for non-family and family businesses, but there is more of a difference in one aspect out of eight—succession management. This conclusion can be considered as a starting point for further directions of possible research of this issue (e.g., the possibility of identifying the reasons for non-succession management in non-family enterprises; analyzing the pros and cons of succession management, based on family or personal relationships in family enterprises,

or the second most common way, which is the management of succession of a rather informal and intuitive nature, based on personal interactions and leadership style), as shown in Tables 3 and 4 (only answers with the highest frequency are listed, however, all answers to all questions from the highest to the lowest frequency were processed).

Since these are collectively micro, small and medium-sized enterprises, the answers to all the questions, with the exception of the absence of succession management for non-family businesses, correspond to the authors' expectations, based on their own experience. Succession management, as a process by which an enterprise determines individuals who will be available both now and in the future to fill certain job roles or provide key competences for the company's future sustainability and growth, is also important in micro, small and medium-sized enterprises and it should, therefore, be given proper attention.

Based on the latest theoretical knowledge of human resources management, the practical experience of the authors and the results of the survey focused on the second area (i.e., used human resources management tools in specific areas of work with people of both non-family and family businesses), certain general recommendations can be formulated in this area, such as using methods of recruitment more actively (i.e., set up succession management system, whether it will be formal or based on other parameters).

### 4.3. Hypotheses Testing

**Hypothesis 1.** *Informal approach in the human resources system/concept is accepted to a greater extent by family businesses than by non-family businesses.*

The first hypothesis is related to the first significant question from Chapter 2. The evaluation was based on the overarching issue of human resources management (i.e., strategy and policy (Do you have a human resources management strategy and policy in your company?)). From the obtained data, the data directly related to the form of strategy and policy were selected (i.e., whether strategy and policy is formally embedded in corporate documents, or whether it is applied informally), based on precedents or the experience of managers or owners, often intuitively or even ad hoc, as shown in Table 9.

**Table 9.** Absolute and relative frequency of answers.

|  | Non-Family Businesses | | Family Businesses | |
|---|---|---|---|---|
|  | **Absolute Frequency** | **Relative Frequency** | **Absolute Frequency** | **Relative Frequency** |
| **Written strategy** | 74 | 18.6% | 40 | 11.6% |
| **Written policy** | 74 | 18.6% | 40 | 11.6% |
| **Informal strategy** | 86 | 21.6% | 68 | 19.8% |
| **Informal policy** | 86 | 21.6% | 68 | 19.8% |
| **None** | 78 | 19.6% | 128 | 37.2% |
| **Total** | 398 | 100 | 344 | 100% |

Source: own elaboration based on data from IBM SPSS Statistics 23.0 (2019).

Often, families create policies without realizing it or being thoughtful about it. These policies are not written but take the form of powerful statements, such as: "That is the way dad (or mom) wanted it" or "That is the way it has always been done around here" [50]. Particularly in the first generation, entrepreneurs may make decisions that they do not think of as policies—they are just seen as decisions. Such decisions can set precedents that become policies. In a sense, the founder embodies the policies. The founders can recognize that their decisions can become precedents for generations to come. In a

family business, where the founder/owner is still in charge, decisions are relatively uncomplicated and typically unchallenged [51].

The first hypothesis was aimed at testing whether the informal approach of family companies in the human resource system also prevails in the presented research. That is, whether they prefer unwritten rules and principles when working with people.

A two-sample comparison test was used. The size of both samples is much larger than 50 and the sample proportions do not reach significantly small or large values. If the null hypothesis is valid, it can be assumed that the statistic test has an asymptotically normalized normal Z distribution.

$$Z = \frac{p_1 - p_2}{\sqrt{p^* \times (1 - p^*) \times \left(\frac{1}{n_1} + \frac{1}{n_2}\right)}},$$

$$p^* = \frac{n_1 \, p_1 \, + n_2 p_2}{n_1 + n_2}$$

where,

$n_{1,2}$—sample sizes (i.e., number of all responses),

$p_{1,2}$—relative frequency of positive responses (i.e., those that prefer informal approach).

Critical field (W) at the significance level ($\alpha$) 0.05, when the null hypothesis is rejected, and the alternative hypothesis is accepted with probability (1—$\alpha$):

$$W = (-\infty, -Z_{0,95}); \text{ i.e., } W = (-\infty, -1{,}645)$$

$$Z = \frac{0.3953 - 0.4322}{\sqrt{0.4151 \times (1 - 0.4151) \times \left(\frac{1}{344} + \frac{1}{398}\right)}}$$

$$Z = -1.019$$

From the aforementioned data, it can be concluded that: $Z \notin W$ (i.e., hypothesis cannot be rejected). The hypothesis that family businesses accept an informal approach in the human resource system/concept more than non-family businesses is accepted. With this finding, the presented research leans towards statements regarding the rather informal environment of a family business, which is also reflected in the field of human resources management. However, policy development cannot be put off forever. Sooner or later, every family business needs to formalize and set down on paper the structure and guidelines under which the family will own and operate their business. Why? The simplest and quickest answer is that, having agreed-upon policies in place and abiding by them, reduce the chances that family conflicts will destroy family business. The family that wants to extend its business to the next generation and generations beyond will find it necessary and useful to move beyond the causal and implicit setting of policy and to deliberately and explicitly create guidelines that chart a successful course for the family and the business [51].

**Hypothesis 2.** *Even in a company where the informal concept of human resources management predominates, formal employee appraisal will be used in more than 66% of businesses.*

The second hypothesis is related to the informal environment, policies and strategic decision-making in the field of human resources management. This also reflects a second research question concerning, inter alia, the nature of the tools used in working with people. From the analyzed areas in Section 4.2. the authors chose tools used in the field of performance management for further research in the area of performance management, as shown in Table 10.

**Table 10.** Absolute frequency of answers pertaining to the character of employee performance appraisal.

|  | **Non-Family Businesses** | **Family Businesses** |
| :---: | :---: | :---: |
| **Informal appraisal** | 142 | 136 |
| **Formal, systematic** | 130 | 124 |
| **Family, personal** | 8 | 16 |
| **None** | 8 | 8 |

Source: own elaboration based on data from IBM SPSS Statistics 23.0 (2019).

This area was chosen because of its direct relationship with costs (employee remuneration) and thus a direct impact on the economic result. The authors wondered whether, in an environment where there are no formal rules for human resources management policy and strategy, formal rules and tools for evaluating employee performance will be created (i.e., in an area that clearly affects profit generation). Does economic responsibility outweigh informality? If the company has set codes and formal rules (of course, to a reasonable extent), it is easier to control, regulate and evaluate the outputs. Researchers assume that, even in companies where the informal concept of human resources management predominates, more than 66% will use a formal appraisal of employee performance. These claims have been applied for both family and non-family businesses.

The relative sensitivity test was used to confirm or rule out the hypothesis. Test criterion for the relative sensitivity test U:

$$u = \frac{\frac{m}{n} - p_0}{\sqrt{\frac{p_0(1-p_0)}{n}}}$$

where,

$n$—number of organizations preferring an informal approach to human resources management concept,

$m$—number of organizations preferring a formal approach for appraisal,

$p_0$—relative frequency.

The significance level is $\alpha = 0.05$, when critical field W$\alpha$ is equal to W 0.05 = {U ≥ u 0.95} = {U ≥ 1.645}, leading to the tested assumption that more than 2/3 (i.e., 66%) of companies with an informal approach to the human resources management concept will apply formal tools for employee appraisal.

Non-family businesses:

$$u = \frac{0.756 - 0.66}{\sqrt{\frac{0.66(1-0.66)}{172}}} = 2.66,$$

relations apply that {U ≥ 1.645}, resp. {2.66 > 1.645}.

The assumption that more than two thirds of non-family businesses, which prefer the informal overall structure of the human resources management system, use formal tools to evaluate employee performance, is accepted. The authors also came to the same conclusion when testing family businesses where the relationship was valid {6.14 > 1.645}. The evaluation of employee performance and related compensation is an emotional topic for both business owners and employees. Organizations are constantly trying to balance the bottom line with paying their workers fairly. That balancing act can become more difficult in a family business if the evaluation of employee performance or compensation strategy is not clearly defined. Because of their very nature, family firms can be complex and, in reality, care needs to be exercised when determining remuneration for all staff, as undoubtedly there are going to be challenges with non-family employees if everyone is not dealt with in a similar manner.

On the basis of the aforementioned data, the answer to research question (3) is: For both types of companies, even if there is an informal concept of the whole human resources management concept, in the area that directly affects the financial management of the company, clear formalized rules prevail. This finding demonstrates a responsible approach to clearly linking performance and remuneration in

the overall performance of the company. It also gives a positive report on family businesses, which seek to prevent conflicts between employees—family members—through formal rules.

## 5. Conclusions

This study compares the human resources management of non-family and family businesses in the Czech Republic. The theoretical basis of this study is based on the literature, which is extremely extensive on this issue. Foreign literature, in particular, from the point of view of research in the Czech Republic, is very rich—Czech literature dealing with this issue is very limited. Therefore, it was drawn mainly from foreign sources.

The research methodology was based on a positivist–objectivist approach, using the quantitative method of a questionnaire survey as the main method and the focus group qualitative research method as the verification method.

The first step in solving the scientific problem was thorough a vast literature review, which resulted in the purpose of the study, which was to identify differences in human resources management in non-family and family businesses operating in the Czech business environment. The stated goal resulted in the formulation of three research questions: (1) Are the overall concept of human resources management different for non-family and family businesses?; (2) Do non-family businesses use tools other than family businesses within the survey population for active work with people (e.g., training, development, motivation, remuneration, etc.)?; (3) Does the informal approach in the field of human resources management prevail in family business? This also led to the formulation of two hypotheses. Subsequently, in June 2019, a survey was conducted in the form of an online questionnaire survey based on a structured questionnaire (verified by the Focus Group method). The basic set included 12,632 enterprises of the manufacturing industry in the Moravian–Silesian region, where the return was 4.6% (a total of 572 questionnaires were returned, 288 from non-family and 284 from family enterprises). The results of the survey were processed and evaluated using IBM SPSS Statistics 23.0 software. During the discussion, the results of the survey were analyzed and commented on in detail, from which more general conclusions were drawn in the researched area, which served as a starting point for formulating further directions of possible research on this issue, which are outlined below. Finally, a two-sample comparison test was performed, which tested the hypotheses of the informality of the human resources management concept by the relative sensitivity test.

The results of the questionnaire survey and hypotheses testing answered the three formulated research questions in the following way. The answer to research question (1) is: The overall concept of human resources management is basically not different for non-family and family businesses; differences can only be found in two aspects out of seven—in human resources management strategy and policy design and in the administration of human resources management. While non-family businesses, in most cases (i.e., 86 (29.9%)), have a strategy and policy, but not in writing, family businesses, in most cases (i.e., 128 (45.1%)), have neither a strategy nor a human resources management policy. The administrative aspect of human resources management in non-family enterprises is, in most cases (i.e., in 124 (43.1%)), ensured by the HR department/personnel officer, and in family enterprises, in most cases (i.e., in 118 (41.6%)) by the owner/owners. The answer to research question (2) is: The tools used in specific areas of work with people are hardly different for non-family and family businesses, but there is a difference in one aspect out of eight—succession management. The non-family businesses most often do not manage succession, but consider introducing a formal system (112; 38.9%), while in family businesses, succession is managed on the basis of family or personal relationships (118; 41.6%). The answer to research question (3) is: For both types of companies, even if there is an informal concept of the whole human resources management concept, in the area that directly affects the financial management of the company, clear formalized rules prevail. The hypotheses did not rule out the claim that there was more informality in the family business environment, unlike non-family businesses. However, in an area that has a clear direct link to the company's performance (i.e., the appraisal of employee performance), both types prefer clearly defined rules.

Since these are collectively micro, small and medium-sized enterprises, the answers to most questions, with the exception of questions pertaining to the well-thought-out and prepared conception of personnel activities and questions pertaining to succession management (for non-family businesses), correspond with the authors' expectations. The results obtained by the survey cannot be compared with any similar survey, as none have been carried out in the Czech Republic so far.

The limiting conditions of the survey can be seen in the return rate of the questionnaires sent, which was 4.6%. The non-existent definition of a family business is also a limitation. The features that have been set for the family business definition for this research are not accurate. The same surname does not mean that they are members of the same family and, vice versa, members of the same family can be called by different names. This risk had to be accepted. Generalizing the findings of this study must be taken with care, as the findings are based on a one-sectional sample of family SMEs in one country. Future research can build on the findings of this research (which can be considered as a pilot research) with studies on larger samples.

In addition to further research on the issue outlined in Section 3 of this study, the authors see another direction of possible research in a detailed analysis of results depending on the "size" of the enterprise (i.e., micro, small and medium-sized enterprise) and also the comparison of the whole human resources management conception and the use of tools in specific areas of work with people not only in non-family and family businesses of different sizes (i.e., micro-enterprises, small and medium-sized enterprises and large enterprises) but also, for example, according to the length of time that the company has been operational.

**Author Contributions:** Conceptualization, P.H.; methodology M.M. and K.K.; validation M.M.; formal analysis M.M.; investigation, P.H. and K.K.; resources P.H.; data curation M.M. and P.H.; writing—original draft preparation P.H.; writing—review and editing, P.H.; visualization, P.H.; supervision M.M.; project administration P.H. and K.K. All authors have read and agreed to the published version of the manuscript.

**Funding:** This research received no external funding.

**Acknowledgments:** The paper was supported within the project of the Student Grant Competition at the Faculty of Economic VŠB-Technical University of Ostrava SP2020/33.

**Conflicts of Interest:** The authors declare no conflict of interest. The funders had no role in the design of the study; in the collection, analyses or interpretation of data; in the writing of the manuscript or in the decision to publish the results.

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
