# Peer review of "Comparison of Human Resources Management in Non-Family and Family Businesses: Case Study of the Czech Republic"

_sustainability, doi:10.3390/su12145493_

Round 1
Reviewer 1 Report
The manuscript “Comparison of human resources management in non family and family business: case study of the Czech Republic” is devoted to the actual scientific problem, namely sustainable entrepreneurship development. The reviewed article is interesting for scholars and theme of the article meets the scope of the journal. Work is performed at sufficient scientific level and has good quality; the results of study human resources management in Czech Republic are professionally interpreted. Prior publication of this manuscript following points need to be addressed:
- References list should be carefully checked and journal style policy should be strictly followed (Abbreviated Journal Name, doi, citation rule for books and monographs, etc). For example, reference 33. Citing through Researchgate is not entirely acceptable.
My decision is minor revision
Author Response
Dear reviewer,
thank you very much for the review and your comment. Based on your recommendation we carefully checked the References and we corrected the errors to strictly follow the journal style policy (we used MDPI_references_guide_v5).
Best regards,
Petra Horváthová, on behalf of all authors

Reviewer 2 Report
Dear Authors,
thank you for improving the manuscript, but I have some comments:
- The abstract of your article is very overloaded with information that could be very well transferred to the Introduction section or methodology.
- In my opinion, the information's which you are given in the line - 63-68 are not needed or have to be rewritten in a more appropriate way.
- And please pay attention to the conclusions: please write it in a more scientific way.
Author Response
Dear reviewer,
thank you very much for the review and challenging remarks. Based on your recommendations we made the following amendments:
- Based on your recommendation concerning the editing of Abstract we:
- moved first two sentences from lines 15 to 18 behind the first sentence in the second paragraph in Introduction (line 46 to 50);
- the word “aim” on line 20 was replaced by the word “purpose” (line 21) and word “article” was replaced by the word “study” (line 21);
- from the sentence „In this context, the authors formulated three research questions and two hypotheses“ (line 23) we removed the introductory part „In this context,…“ ;
- sentence „Primary data, obtained through a quantitative survey (using a questionnaire) from a total of 572 respondents (288 non-family and 284 family businesses), were processed using IBM SPSS Statistics software.“ we removed from the Abstract (lines 24 to 26); this re-formulated sentence also replaced the sentence on line 235 in Chapter Methods and Data (line 236);
- we added second to last sentence: „The article's main findings are, non-family and family business do not substantially differ in human resource management.“ (lines 26 a 27);
- last sentence of Abstract (originally lines 27 to 30) we shortened to: The article is formulating more general conclusions in the researched area, which can serve as a starting point for further direction of possible research on this issue (lines 28 to 30).
- Based on your recommendation concerning the information's which were given in the original lines 63-68, we removed those information from these lines (lines 68 to 73).
- Based on your recommendation concerning Conclusions, we removed the first two paragraphs (lines 571 to 607) and we replaced them by two different paragraphs which were created by the reformulation and addition of the original (removed) paragraphs. Remaining lines 655 to 684 stayed unchanged.
- Also, the whole article has been proof read again, and some minor typos have been corrected, i.e. missing full stop, missing subject, etc.
Best regards,
Petra Horváthová, on behalf of all authors

Reviewer 3 Report
I took note of the improvements made to the article sustainability-857808 is resubmitted from sustainability-817188.
After a careful check of those sent in the letter below, I declare myself satisfied.
As a result, I have no other comments.
I assure you of all my consideration.
He received the following letter through the publisher:
”Based on the remarks and recommendations of the second reviewer we have added the following 2 sentences to the first paragraph of the Introduction, we have modified one sentence:
The basic strategic objectives of each organization include long-term growth and sustainability [1]. Human resources and their management play a critical role in the successful operation of any company [5]. Practical human resources management, which affects the well-being of employees and thus the performance of the company [6], a which includes a wide range of activities (e.g. development of personnel strategy and policy, ways of recruiting and selecting, evaluating, remunerating or promoting employees, etc.), may vary among businesses, depending on whether it is a family or non-family business.
We also added the following paragraphs in the Introduction:
We can state that research concentrating on family businesses is largely overlooked in the Czech Republic. More extensive research was carried out at TU Liberec, VUT Brno with the output of the book [7] and at the University of Economics in Prague with the results presented in the articles [8,9,10,11]. Impact research is carried out by the Association of Small and Medium-sized Enterprises. However, overall, the research does not correspond to the intensity and scope devoted to this topic on a global scale. The issue of family business is very important in terms of the development of the business environment and GDP creation. Family businesses are therefore crucial for economic progress [12, 13].
The following paragraph has been added to the Methods and Data chapter:
This method was chosen for its advantages over other options for addressing respondents. The advantages of this method are the ability to address a large number of respondents and at the same time address those who are not willing to provide a personal interview, as well as the elimination of the relationship between respondent and researcher, which may allow obtaining a relatively exact and objective opinion. However, the questions must be precise and comprehensible so that they are clearly understood and do not require further explanation. Other advantages include, for example, low costs, high return, speed of answers, versatility and greater sincerity of respondents than, for example, in personal interviews. On the contrary, the risks of this method include the fact that not everyone is connected to the Internet and also the susceptibility to technological problems [53].
From Conclusion the lines no. 621-625 were removed.
In References negligence’s in writing were corrected.
Text of the article has been revised by English native speaker.
Author Response
Dear reviewer,
thank you very much for the review.
Best regards,
Petra Horváthová on behalf of all authors
This manuscript is a resubmission of an earlier submission. The following is a list of the peer review reports and author responses from that submission.
Round 1
Reviewer 1 Report
The idea from which to start and the sampling have some design problems.
In a family company there can be no objective judgment regarding human resource management. At the conceptual level it is normal that there are no differences, but at the implementation level it is clear that there are. The authors themselves say: The definition of a family business itself is not unambiguous - the definition of a family business varies from one country to another, but it always takes into account the number of family members participating in the company's capital, the amount of equity to be paid by family members, enterprise successor generation, etc. The most important feature is that the business is based on the personality of the founder, on the intensity of family involvement in the business, on the commitment of family members to perform more on their own, on quality, because to represent their own name and tradition is a prerequisite for success in future generations. This fact clearly alters the idea of human resources management when it comes to the fault and its members because the family does not manage human resources.
The questionnaire was not pretested and validated, which can be seen from the answers. The questions refer to different terms and you cannot test 2 concepts in the same question.
The idea of grouping by size in terms of the number of employees is not desirable in such research. Some elements did not need to be tested by questionnaire because they are "axiomatic" given the specifics of the companies surveyed.
While large enterprises carry out staffing activities on a continuous basis, in micro, small and medium-sized enterprises, some staffing activities are carried out only occasionally.
- Do you have a human resources management strategy and policy in your company?
Strategy is a comprehensive plan, made to accomplish the organizational goals.
Policy is the guiding principle, that helps the organization to take logical decisions. In business parlance, the terms strategy refers to is a unique plan designed with the aim of achieving a competitive position in the market and also to reach the organisational goals and objectives. In short, it is an interpretative plan, that guides the enterprise in realizing its goal. On the other hand, policy refers to a set of rules made by the organisation for rational decision making.
Policy lays down the course of action, which is opted to guide the organization’s current and future decisions. Many people have confusion regarding the two terms, but they are not alike.
- Does the business owner(s) have the necessary knowledge and skills in human resources management?
Two words that describe a person’s competence knowledge and skill! At first glance, both of them seem synonymous but give it some thought and you would realize both of them are very different concepts. Knowledge refers to learning concepts, principles and information regarding a particular subject(s) by a person through books, media, encyclopedias, academic institutions and other sources. Skill refers to the ability of using that information and applying it in a context. In other words, knowledge refers to theory and skill refers to successfully applying that theory in practice and getting expected results.
The very low response rate raises questions about the validity and generalization of the research.
Reviewer 2 Report
The work "Comparison of human resources management in non-family and family business: case study of the Czech Republic" it is important from several points of view. The author in the Introduction also refers to some of these.
I appreciate it is well structured (work).
However, out of the 43 bibliographic references, 39 are confined to the “Literature Review” section, and 3 to the “Introduction” section (of which 2 - self-quotations).
I recommend expanding them with several recent articles (2018-2020), published in WoS journals.
The method used and the data reflect an obvious scientific character.
However, I think the advantages and risks of the chosen method must be explained (is it infallible?).
From the "Conclusions" the lines no. 621-625, which repeats some ideas (unnecessarily).
”References” present some negligence in the writing.
Exemple:
- BaÅŸkurt, G.; AltindaÄŸ, E. The Impact of Institutionalization of Family Business on Strategic Human Resources Management and Company Performance. Business Management Dynamics 2017, 7(3), 10-25. Retrieved August, 30, 2019 Available online: http://bmdynamics.com/issue_pdf/bmd110926-10-25.pdf (accessed on 30 August 2019).
You can skip playing long links (which take up 3 rows):
- Sharma, E. (2012). HR issues and intervention model for family business. International Journal of Business Economics & Management Research 2012, 2(12), 288-297. Available online: https://www.researchgate.net/profile/Drekta_Sharma/publication/262909110_HR_ISSUES_AND_INTERVENTION_MODEL_FOR_FAMILY_BUSINESS/links/0c9605391f3f4d4754000000/HR-ISSUES-AND-719 INTERVENTION-MODEL-FOR-FAMILY-BUSINESS.pdf (accessed on 9 September 2019).
The system of bibliographic references must be arranged according to the rules of publication (Sustainability ISSN:2071-1050).
All in all, eventually, it may be seen by an English teacher (native).
Reviewer 3 Report
Dear Authors,
Thank you for the possibility to read your manuscript.
The paper is well prepared and the writing style is on the high scientific level. The cited literature is adequate and up-to-date.
The used methodology is appropriate and enabled authors to obtained adequate scientific results.
I suggest to the authors to improve the conclusion and insert the comparison of the obtained results and that one from the similar previous researches.
Reviewer 4 Report
The text is coherent, analyzes are detailed, and observations are carefully described.
Reviewer 5 Report
The manuscript “Comparison of human resources management in non family and family business: case study of the Czech Republic” is devoted to the actual scientific problem, namely sustainable entrepreneurship development. The reviewed article is interesting for scholars and theme of the article meets the scope of the journal. Work is performed at sufficient scientific level and has good quality; the results of study human resources management in Czech Republic are professionally interpreted. The manuscript may be considered for publication after minor revision in Sustainability. Prior publication of this manuscript following points needs to be addressed:
- For a better perception of the material, some tables should be presented in the form of diagrams or figures. I mean tables 1-3, 7,8. Although I do not insist on his remark and leave it to the authors.
- In the conclusions it is necessary to remove the general phrases. For example: "The aim of the article was to identify differences in the overall concept of human ...", "The importance of family business for the development of the business ...", etc. Conclusion must be more focused on results of research. Moreover, It would be good to add a more detailed presentation of ways to resolve the scientific problem.
- Moderate English changes required. There are grammar/typing and orthographical errors in the manuscrip
My decision is minor revision